ecology

island biogeography, spatial subsidies, marine-derived nutrients, avian ecology

**Author for correspondence:**
Debora S. Obrist
e-mail: dobrist@sfu.ca

# Marine subsidies mediate patterns in avian island biogeography

Debora S. Obrist[1,2], Patrick J. Hanly[1,2], Jeremiah C. Kennedy[1,2], Owen T. Fitzpatrick[2,3], Sara B. Wickham[2,3], Christopher M. Ernst[1,2], Wiebe Nijland[2,3,5], Luba Y. Reshitnyk[2], Chris T. Darimont[2,4,6], Brian M. Starzomski[2,3] and John D. Reynolds[1,2]

[1]Earth to Ocean Research Group, Department of Biological Sciences, Simon Fraser University, 8888 University Drive, Burnaby, British Columbia, Canada V5A 1S6
[2]Hakai Institute, PO Box 309, Heriot Bay, British Columbia, Canada V0P 1H0
[3]School of Environmental Studies, and [4]Department of Geography, University of Victoria, 3800 Finnerty Road, Victoria, British Columbia, Canada V8P 5C2
[5]Department of Physical Geography, Utrecht University, Princetonlaan 8a, 3584 CB Utrecht, The Netherlands
[6]Raincoast Conservation Foundation, PO Box 2429, Sidney, British Columbia, Canada V8L 3Y3

DSO, 0000-0002-5645-6037; JCK, 0000-0003-0072-6840; OTF, 0000-0001-7715-3601; SBW, 0000-0001-8155-5689; CME, 0000-0002-0257-6252; WN, 0000-0002-2665-0947; CTD, 0000-0002-2799-6894; BMS, 0000-0001-5017-5405; JDR, 0000-0002-0459-0074

The classical *theory of island biogeography,* which predicts species richness using island area and isolation, has been expanded to include contributions from marine subsidies, i.e. *subsidized island biogeography* (SIB) *theory.* We tested the effects of marine subsidies on species diversity and population density on productive temperate islands, evaluating SIB predictions previously untested at comparable scales and subsidy levels. We found that the diversity of terrestrial breeding bird communities on 91 small islands (approx. 0.0001–3 km$^2$) along the Central Coast of British Columbia, Canada were correlated most strongly with island area, but also with marine subsidies. Species richness increased and population density decreased with island area, but isolation had no measurable influence. Species richness was negatively correlated with marine subsidy, measured as forest-edge soil $\delta^{15}$N. Density, however, was higher on islands with higher marine subsidy, and a negative interaction between area and subsidy indicates that this effect is stronger on smaller islands, offering some support for SIB. Our study emphasizes how subsidies from the sea can shape diversity patterns on islands and can even exceed the importance of isolation in determining species richness and densities of terrestrial biota.

## 1. Introduction

The *theory of island biogeography* (TIB) [1] predicts that species richness on islands is driven by an immigration rate, determined by island isolation and an extinction rate, which depends on island size. TIB has been expanded to consider other factors that mediate the effects of area and isolation, including speciation [2], variability in primary productivity [3], climate [4], exposure to prevailing winds and ocean currents [5], habitat diversity [6,7], invasive species [8], and spatial subsidies [9–12].

Nutrient-limited habitats often receive subsidies from adjacent ecosystems [13]. Nutrient transfers can be controlled by the donor habitat or result from foraging by recipient-habitat species [14], and can have large impacts on nutrient-limited islands. *Subsidized island biogeography* (SIB) theory predicts that insular species richness will either increase or decrease with subsidy input, depending on where islands lie on a unimodal productivity–diversity curve [15]. While classical TIB assumes constant population densities across islands regardless of size, spatial

subsidies can shift species densities [16–20]. As subsidies increase productivity, islands can support denser populations, reducing extinction risk from demographic stochasticity and thereby increasing expected diversity. Conversely, beyond some productivity threshold, a few species at high densities may outcompete others, leading to lower diversity [15]. SIB posits that these effects are amplified on smaller islands because subsidies have a higher per-unit-area effect on productivity [9].

Since the inception of SIB, studies have shown that the productivity–diversity relationship is probably scale and system dependent [21,22]. SIB has mostly been investigated on dry, arid islands at low latitudes [10,12], with one recent exception that found no influence of marine productivity on island diversity at global or regional scales [11]. No empirical tests have been conducted in more productive temperate island systems, but nutrients from salmon carcasses are established drivers of songbird [16–18,20], invertebrate [23] and plant communities [19] in mainland coastal temperate rainforests. In addition, there is a surprising lack of research on subsidy effects on bird communities, despite the avian focus of many island biogeography studies [1,4,6,7,24–27].

Here, we conducted, to our knowledge, the first empirical test of the effects of marine subsidies on classical TIB predictions on temperate rainforest islands. We quantified marine subsidies and tested their effects on terrestrial breeding bird communities on 91 small islands (less than 3 km²) along the Central Coast of British Columbia, Canada. We used hierarchical models to test the importance of classical TIB predictors (island area and isolation) relative to island-specific predictors of subsidy acquisition and retention (shore-cast macroalgal biomass, shoreline substrate and forest-edge soil nutrients) on avian species richness and density. This approach provides a finer resolution than previous studies where island subsidies were treated as binary (i.e. presence/absence), or where island subsidies were predicted from mainland accumulations [12]. Our study region contains nearly 1600 islands, many of which are too small and topographically simple to host salmon-bearing streams, but they could receive nutrient transfers from foraging bald eagles (*Haliaeetus leucocephalus*) and river otters (*Lontra canadensis*), and from shoreline seaweed deposits. Nearly all surveyed islands had evidence of river otters (greater than 90%), whose activities have been observed to subsidize temperate coastal forests through excrement and food scraps [28,29]. Furthermore, the northeastern Pacific Ocean exhibits some of the highest levels of primary productivity in the world, resulting in large kelp forests [30]. In the Bahamas, shore-cast macroalgal deposits directly increase terrestrial productivity, and the diversity and abundance of shoreline invertebrates and their predators [31].

Consistent with TIB, we predicted that island area would be a strong predictor of terrestrial bird species richness, with a minimal effect on density [1]. We did not expect isolation to have an effect because birds are unlikely to be dispersal-limited on near shore islands [4,6]. We predicted that species richness and density would be higher on heavily subsidized islands, as populations with more individuals have lower extinction rates and higher equilibrium species richness [17,18,20]. Depending on where these islands fall on the unimodal productivity–diversity curve, we also anticipated that additional subsidies could move the system towards an equilibrium with fewer species [9]. We tested the hypothesis that islands with more macroalgal deposition, higher δ¹⁵N in forest-edge soils and greater receptivity to subsidies (less rocky shoreline) would host more species and more birds per-unit-area.

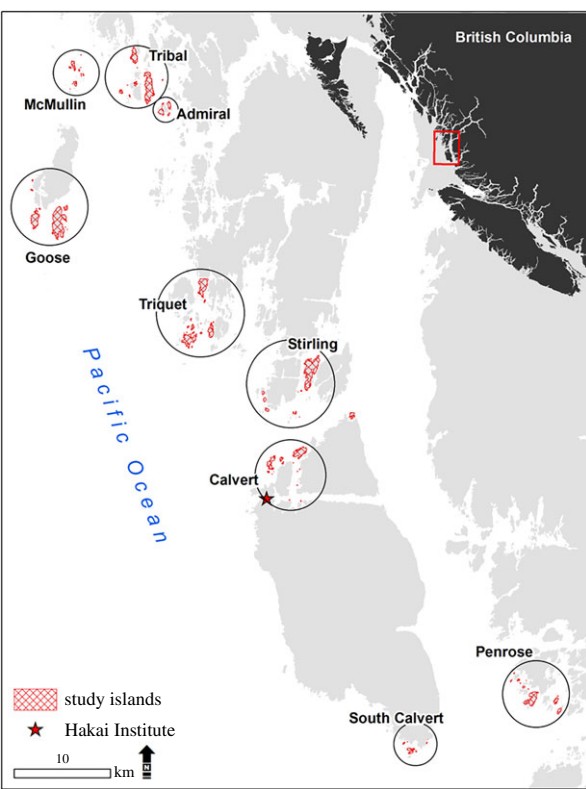

**Figure 1.** Outer islands in the Central Coast region of British Columbia, Canada. Circles surround island groups that were studied. Each group contains between 6 and 17 study islands. (Online version in colour.)

## 2. Material and methods

### (a) Study area

We surveyed 91 islands in the Central Coast region of British Columbia, Canada (figure 1; 51°26′ to 52°3′ N and 127°41′ to 128°28′ W). This area is in the very wet, hyper-maritime subzone of the Coastal Western Hemlock biogeoclimatic zone, which receives over 3000 mm precipitation annually [32]. We selected islands representative of the region's biogeographical and geomorphological variation (electronic supplementary material, table S2) with a maximum area of 3 km² for sampling feasibility. We defined an island as a landmass with terrestrial vegetation that is separated from neighbouring landmasses during high tide.

### (b) Terrestrial breeding birds

We surveyed terrestrial breeding birds on each island with 10 min point count surveys between early May and mid-July in either 2015, 2016 or 2017. We spaced point counts at least 250 m apart to maintain independence [33]. Two surveys were conducted at each location, approximately one month apart, to account for detection differences in early and late season migrant species. To reduce the effects of edge habitat we avoided placing points within 50 m of the shore. Point counts were not conducted during rain or wind speeds above 3 on the Beaufort scale. To minimize inter-observer bias, surveyors were selected based on their ability to identify birds of British Columbia.

### (c) Nutrient subsidies

We considered nutrient subsidies from three factors: (i) the biomass of beach-cast seaweeds (wrack); (ii) the proportion of shoreline classified as 'rocky', a metric inversely related to an island's ability to receive and retain wrack [34]; and

(iii) forest-edge soil $\delta^{15}N$, a composite measure of other marine-derived nutrients vectored by wind, water and animals (e.g. river otter faeces and food scraps, sea spray and seabird guano).

To measure wrack biomass, we placed two 20 m transects parallel to the water at four locations on every island, representing the north, east, south and west-most points. We placed one transect on the most recent high tide line, and one at the most recent storm line—the highest wrack line on the beach. We weighed wrack in three randomly placed 1 m² quadrats along each transect [34]. Wrack wet weights were calibrated to dry weight by species [35]. To linearize the data for analysis at an island level, we used the square root of the mean wrack biomass of the 24 biomass measurements per island.

Shoreline substrate is the most important predictor of wrack retention on islands in this region; wrack accumulates less on rocky shorelines than on other substrates (i.e. boulder, cobble, gravel and sand) [34]. To calculate the proportion of rocky shoreline on each island, we used ESRI ArcMap 10.3 to categorize shoreline substrate at 5 m intervals around each island using raster data collected with small remotely piloted aerial systems (sRPAS) at 10 cm resolution.

We used forest-edge soil $\delta^{15}N$ as a direct measure of marine subsidies before they attenuate throughout the island. Previous studies on these islands showed that soil $\delta^{15}N$ decreases significantly with increased distance from shore (O Fitzpatrick 2018, unpublished data), implying marine origin [36]. Levels of $\delta^{15}N$ also tend to be higher at river otter activity sites [29,37] and near eagle nesting trees (R Miller 2019, unpublished data). Soil samples were taken from vegetated sites adjacent to the shoreline at each of the four cardinal directions on each island and analysed for stable isotopes. To obtain a single measure for each island, we averaged these four $\delta^{15}N$ values. Interestingly, there was a moderate negative correlation between island area and forest-edge soil $\delta^{15}N$, despite all points being adjacent to the shoreline (electronic supplementary material, figure S3).

### (d) Environmental covariates

We measured island area and the normalized difference vegetation index (NDVI) with WorldView-2 satellite imagery. By combining NDVI and visual inspection of sRPAS imagery, we distinguished five habitat categories on our islands: dense/closed-canopy forest, light/open-canopy forest/dense shrub, light shrub/grass, bog (vegetation and water), and woody debris/snags. We determined the relative proportion of these habitats on each island, and calculated the Shannon diversity index to represent heterogeneity [38]. We calculated distance to the nearest island large enough (120 245 m²) to act as a functional 'mainland' to represent isolation (methods in the electronic supplementary material).

### (e) Data analysis

#### (i) Avian species richness

To account for missed detections and unequal numbers of point counts on islands of different sizes, we estimated species richness with Chao1 [39] using the 'vegan' package [40] in R. We only considered birds detected within 50 m of the observer to minimize errors in distance estimation, differences in species detectability and double-counting. To investigate factors influencing species richness on an island-level scale, we fitted a series of linear mixed-effects models (LMMs) representing a suite of hypotheses and evaluated them using Akaike's information criterion, corrected for small sample sizes (AICc). We compared an area-only model with models containing isolation, habitat heterogeneity and both (electronic supplementary material, table S3). Because these parameters did not improve model fit, we dropped them from further analyses.

To assess the relative importance of each remaining variable (RVI) we model-averaged a set of LMMs with all possible subsets of island area, forest-edge soil $\delta^{15}N$, proportion of rocky shoreline and wrack biomass (kg m$^{-2}$) using the 'lme4' package in R [41]. We scaled and centred all independent variables, log-transformed island area and estimated species richness, and square root-transformed wrack biomass to linearize the relationship prior to standardizing. We included interactions between area and both direct subsidy measures: forest-edge soil $\delta^{15}N$ and wrack biomass. Because substrate is the best predictor of shoreline wrack accumulation, we also considered an interaction between proportion of rocky shoreline and wrack biomass. Assuming that rocky shoreline could influence subsidy vectors that affect forest-edge soil $\delta^{15}N$, we considered this interaction as well. We assumed the combined effect of wrack biomass and forest-edge soil $\delta^{15}N$ was additive. All models included year and a random effect of node (a cluster of nearby islands sampled in one survey period).

#### (ii) Avian density

We followed the same procedures to analyse patterns in avian density. First, we used AICc to confirm that generalized linear mixed-effects models (GLMMs) containing isolation, habitat heterogeneity or both were not more informative than a model with area alone (electronic supplementary material, table S4). We then used the candidate models from the species richness analysis to determine the effects of subsidies on avian density. We fitted a series of GLMMs with each island's raw bird abundance (sum of all individuals within 50 m of observers) as the response. The total area surveyed per island was included as an offset term. Because we wanted to predict relative densities to make comparisons across islands, we did not account for variation arising from differences in detectability.

## 3. Results

### (a) Avian species richness

We conducted 566 point counts in 283 locations on 91 islands and detected 32 species of terrestrial breeding birds (electronic supplementary material, table S1). Raw species richness ranged from 0 to 20 species, and estimated species richness (Chao1) ranged from 0 to 30 species.

As predicted, island area was the strongest predictor of terrestrial bird species richness (figure 2a; RVI: 1.00). The effect of the area was nearly 3.5 times stronger than the effect of forest-edge soil $\delta^{15}N$ (figure 2b; RVI: 0.87). Average-sized islands (16 571 m²) are predicted to host 8.23 ± 1.13 (model-averaged estimate ± 95% confidence interval (CI)) species, whereas islands one order of magnitude larger (159 485 m²), or smaller (1722 m²) would host 13.27 ± 1.19 and 5.11 ± 1.18 species, respectively. Forest-edge soil $\delta^{15}N$ had a negative relationship with bird species richness, with approximately one species lost for every 3‰ increase above the mean (6.8‰). All other variables had low RVIs and 95% CIs that overlapped with zero (figure 2c).

### (b) Avian density

Bird densities on islands ranged from 0 to 171 individuals ha$^{-1}$ (median = 19.1; mean = 26.5 ± 5.3). Island area was a strong negative predictor of density (figure 2d; RVI: 0.99). Forest-edge soil $\delta^{15}N$ and the interaction between island area and soil $\delta^{15}N$ were also highly important (RVI: 0.99 and RVI: 0.92, respectively). The effect of island area was nearly twice as strong as that of soil $\delta^{15}N$; however, the effect of $\delta^{15}N$ was not significant owing to the large uncertainty (figure 2f). The interaction between island area and

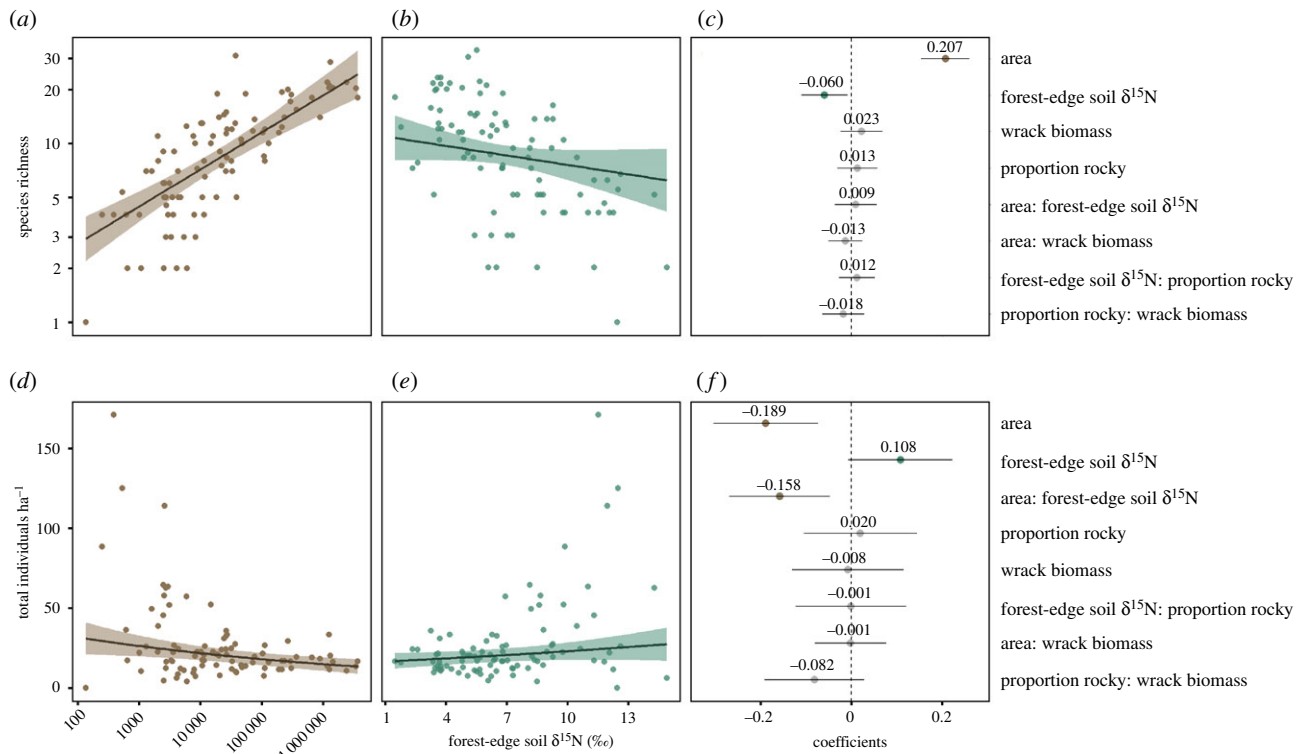

**Figure 2.** Model-averaged predictions for species richness (*a,b*) and bird density (*d,e*) as a function of the area and forest-edge soil $\delta^{15}$N, holding all other variables constant at their observed means. Shaded areas represent 95% confidence intervals. Model-averaged, standardized coefficients are ranked in descending order of their relative variable importance (RVI) (*c,f*). For species richness (*c*), the RVIs are: area (1.00), forest-edge soil $\delta^{15}$N (0.87), all others (less than 0.43). For density (*f*), the RVIs are: area (0.99), forest-edge soil $\delta^{15}$N (0.99), area × forest-edge soil $\delta^{15}$N (0.92), all others (less than 0.33). Error bar represents 95% confidence interval of model-averaged coefficient estimate. (Online version in colour.)

soil $\delta^{15}$N was significant, and nearly as strong as the effect of island area itself. The interaction coefficient was negative indicating that this marine subsidy effect decreased with island area. An average-sized island is predicted to host $20.7 \pm 4.1$ individuals ha$^{-1}$ versus $17.2 \pm 4.0$ and $25.0 \pm 5.8$ individuals ha$^{-1}$ for islands one order of magnitude larger and smaller, respectively. Forest-edge soil $\delta^{15}$N had a positive relationship with bird relative density (figure 2*e*), with a single standard deviation increase (approx. 3‰) above the mean resulting in an increase of over 2 individuals ha$^{-1}$ $(23.0 \pm 5.5)$.

## 4. Discussion

We explored how cross-ecosystem spatial subsidies mediate classical island biogeography predictions for terrestrial breeding bird species richness and population density on 91 islands on the Central Coast of British Columbia, Canada. In accordance with classical TIB [1], island area was the most important predictor of terrestrial bird species richness (figure 2*a*). We found that nutrient subsidies affect this relationship: islands heavily subsidized by marine nutrients had fewer species (figure 2*b*). Accordingly, our study system cannot be reduced to the two classical TIB predictors: isolation was not an important predictor of bird species richness, and one measure of marine subsidy (forest-edge soil $\delta^{15}$N) was the second most important predictor of bird species richness. In addition, while TIB makes no predictions about how population density varies with island area, we found lower

densities of birds on larger islands, and higher densities on more subsidized islands (figure 2*d,e*). We found that the effect of subsidies on density was stronger on smaller islands.

Overall, we found little support for SIB theory as a stand-alone predictive framework for understanding the influence of subsidies on bird species richness. One of the key predictions of this theory is that smaller islands are disproportionately affected by marine subsides owing to a larger perimeter-to-area ratio. A larger interface between land and sea should increase the potential for marine input, leading to increased terrestrial productivity, higher population densities and lower extinction rates, with the potential to affect species diversity [9,15]. In our study, species richness on smaller islands was not disproportionately affected by marine subsidies, but smaller, more subsidized islands hosted more dense populations. Because these islands are relatively productive, one possibility is that additional nutrients push the community towards the downward-sloping side of the hypothetical unimodal productivity–diversity curve. Theoretically, higher productivity could increase species richness up to a certain threshold, beyond which species richness decreases owing to exclusion by competitively dominant species [42]. Although this relationship has received strong support in early productivity-plant-diversity literature [43] and mixed support in productivity-animal-diversity literature [22,44], more recent re-analyses of productivity–diversity relationships suggests that unimodal and negative relationships between diversity and productivity are extremely rare in both plants and animals, at all spatial scales [21].

Our results differ from similar studies in the Pacific northwest, where nutrients derived from Pacific salmon are associated with higher species richness and population densities at mainland sites [18,20]. This is probably owing to the source of subsidy. While the underlying mechanism is the same, (i.e. soils are enriched and stimulate plant productivity), salmon provide a large, predictable flux of nutrients without disturbing coastal vegetation or increasing songbird predation risk. Given their ubiquity in our system, is likely that $\delta^{15}N$ enrichment in forest-edge soils results from river otter activity, which is more variable in intensity and duration [45]. Although river otter faeces can directly fertilize riparian plant communities [28], otters could drive behavioural changes as birds actively avoid direct predation and/or nest predation. Alternatively, otters can reduce the structural complexity of coastal vegetation [29], an important predictor of bird diversity [46]. An additional contribution to the observed pattern may relate to marine exposure. Plant communities on many of our islands reflect their windswept, salt-sprayed environment with dwarfed trees and dense shrub thickets [47]. Marine nutrients are deposited on land through marine fog, wind and sea spray, which may raise soil $\delta^{15}N$ while creating unfavourable conditions for many bird species. We also considered that certain species/feeding guilds could be driving patterns in our study; our data suggest that the insectivore guild may be important but there is too much uncertainty to draw meaningful conclusions (electronic supplementary material, figure S4).

Habitat diversity was not related to island-scale avian diversity, despite contrary findings elsewhere at a range of spatial scales [7,27,46,48]. Either our coarse categorizations did not capture habitat components that underlie avian habitat affinities, or our species pool is not particularly sensitive to habitat variation. Habitat diversity is also unimportant for arboreal arthropods in Florida mangroves [49], snakes on eastern Nearctic islands [50] and vascular plants on Swedish islands [51].

Finally, we found no support for the prediction that wrack biomass or shoreline substrate affect bird species richness or density on islands. Wrack decays in 1–30 days [52], and wrack deposition in this region is significantly lower in July than during winter [34]; the wrack biomasses we recorded may not represent annual or seasonal input. It is also possible that any subsidizing effects of wrack and shoreline substrate are dwarfed by the effects of river otters and other unmeasured sources (e.g. fog and sea spray) contributing to soil $\delta^{15}N$.

Subsidies may not directly increase productivity in our system. Soils are nitrogen limited; we observed total nitrogen levels of 0.9–2.4‰, which is comparable to those of other nitrogen-limited forests in the Pacific northwest (0.7–3.8‰). We observed a greater $\delta^{15}N$ range on islands compared to mainland coastal forests (1.2 to 15.3‰ versus −2.9 to 6‰, respectively [53–57]), which suggests that insular soil nitrogen is largely marine derived. Islands with higher $\delta^{15}N$ also had higher levels of total nitrogen ($R^2 = 0.08$, $p < 0.001$). However, the $\delta^{15}N$ signal is noisy, and levels can also be affected by soil fractionation processes.

We provide a novel test of SIB on avian diversity on 91 islands that vary in area and levels of marine subsidy. We show that marine subsidies increase avian density but decrease species richness in a productive temperate island ecosystem. These results suggest that marine subsidies can cascade to higher trophic levels and alter diversity in a manner infrequently observed for animals (i.e. a negative relationship between productivity and diversity). In agreement with SIB, we find that small island communities are more strongly influenced by subsidies. While these results neither prove nor disprove the subsidy-richness relationship proposed by SIB [9], they confirm that subsidies play an important role in shaping insular biodiversity, even relative to biogeographic variables like isolation and habitat diversity.

Ethics. This study was conducted with permission from both Haíɫzaqv (Heiltsuk) and Wuikinuxv governments. It was also conducted in British Columbia Provincial Protected Areas within the Hakai Lúxvbálís Conservancy, the Calvert Island Conservancy, the Outer Central Coast Islands Conservancy, the Penrose Island Marine Provincial Park and the Penrose-Ripon Conservancy under BC Parks Permit no. 107190. Avian point count surveys were conducted under SFU Animal Care Permit no. 1165B-15.

Data accessibility. Data for this project are archived at the Hakai Institute, and can be accessed at hecate.hakai.org. Search the Metadata Catalogue for 'Avian and paired vegetation data from 100 Islands Project (Central Coast)—2015–2017.' Code for analyses is available from github.com/debobrist/avian-sib/.

Competing interests. We declare we have no competing interests.

Funding. This work was supported by the Tula Foundation at the Hakai Institute, MITACS, NSERC Discovery Grants to J.D.R., B.M.S. and C.T.D., and an NSERC CGS-M to D.S.O.

Acknowledgements. We thank the Haíɫzaqv (Heiltsuk) and Wuikinuxv First Nations for their support in conducting this research. We are deeply grateful to the Hakai Institute for field and laboratory support, and to the multitude of field and laboratory technicians who made this work possible. Thank you also to Carl Humchitt for your guidance and knowledge.

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
