## [Reviewer comments · Proceedings of the Royal Society B: Biological Sciences]

Review History

RSPB-2019-1889.R0 (Original submission)

Review form: Reviewer 1

Recommendation

Major revision is needed (please make suggestions in comments)

Scientific importance: Is the manuscript an original and important contribution to its field?

Good

General interest: Is the paper of sufficient general interest?

Good

Quality of the paper: Is the overall quality of the paper suitable?

Good

Is the length of the paper justified?

Yes

Should the paper be seen by a specialist statistical reviewer?

No

Do you have any concerns about statistical analyses in this paper? If so, please specify them explicitly in your report.

No

It is a condition of publication that authors make their supporting data, code and materials available - either as supplementary material or hosted in an external repository. Please rate, if applicable, the supporting data on the following criteria.

Is it accessible?

Yes

Is it clear?

Yes

Is it adequate?

Yes

Do you have any ethical concerns with this paper?

Yes

Comments to the Author

This is an interesting and well-written paper which tests a hypothesis on a new twist to the classical equilibrium theory of island biogeography; the effect of marine subsidies on the species area curve. The hypothesis incorporates the proposed hump-shaped relationship between species richness and productivity, making the idea captivating. Previous studies of plants have found evidence for this hypothesis, but similar studies of mobile animals do not exist to my knowledge, making this study unique. The research team is highly qualified for this investigation and their analytical methods are superb. The results which provide some support for the hypothesis are intriguing; terrestrial bird species richness decreased with increasing marine subsidies. However, the conclusion on the mechanism of this effect is quite tentative for several reasons.

The basic premise is that productivity of the study system is high which is on the declining side of diversity-productivity curve. Although previous studies have found that productivity is high on the mainland of British Columbia, productivity on the small study islands may have been substantially lower, and thereby on the increasing side of the diversity-productivity curve. If so, according to the hypothesis, the relationship between species richness and marine subsidies should be positive, not negative as found in this study.

Previous studies cited in this paper on the mainland of British Columbia found that salmon-derived nutrients increased diversity of terrestrial bird species (Field, R.D. & Reynolds, J.D. 2011, Wagner, M.A. & Reynolds, J.D. 2019), but no explanation is given for the conflicting results of those studies and this study. This needs to be addressed.

In accordance with the subsidized island biogeography theory, the authors suggest that increasing productivity via marine subsidies increased overall bird density but decreased species richness due to competitive exclusion by dominant species. However, there is no evidence for competition in the paper. Data on species presence/absence and abundances on islands with high and low subsidies may shed some light on competitive interactions. As it stands now, support for the subsidized island biogeography theory is weak. I must commend the authors for including alternative hypotheses. Predators associated with marine subsidies, such as otters, eagles and sea birds (particularly seagulls) may exclude certain highly vulnerable terrestrial bird species but not others that are less vulnerable to predation. The other alternative hypothesis is that high inputs of marine subsidies may occur on islands with habitats that are not suitable for some bird species.

This phenomenon may be common in other systems as well. Data on the association between habitat types and marine subsidies would be helpful. I feel that the alternative hypotheses are quite plausible and deserve more attention.

The Abstract and Results state the positive effect of subsidies on density is stronger on smaller islands but there is no reference to a test for this pattern. Is it the interaction between area and forest edge subsidies? A plot of area vs forest edge subsidies would be helpful.

Review form: Reviewer 2

Recommendation

Major revision is needed (please make suggestions in comments)

Scientific importance: Is the manuscript an original and important contribution to its field?

Excellent

General interest: Is the paper of sufficient general interest?

Excellent

Quality of the paper: Is the overall quality of the paper suitable?

Good

Is the length of the paper justified?

Yes

Should the paper be seen by a specialist statistical reviewer?

No

Do you have any concerns about statistical analyses in this paper? If so, please specify them explicitly in your report.

No

It is a condition of publication that authors make their supporting data, code and materials available - either as supplementary material or hosted in an external repository. Please rate, if applicable, the supporting data on the following criteria.

Is it accessible?

Yes

Is it clear?

Yes

Is it adequate?

Yes

Do you have any ethical concerns with this paper?

No

Comments to the Author

Obrist et al. assess how marine subsidies influence island biogeography patterns in birds, with a well replicated study in western Canada. This is a nice contribution, and will add a robust data-rich example to this debate. I do have some issues for the authors to address.

It is not clear enough in the manuscript if your response (the bird data) are all terrestrial bird species, or if it includes some marine birds. In the methods you note that seabird guano is one of the contributing inputs for the marine subsidies. It is important to be clear if your response is just terrestrial birds, that feed on land, to remove the potential for circularity in the narrative / analysis (i.e. the bird numbers meaning more marine D15N signal in the soil). If your response does include both terrestrial and marine birds, I suggest you do a supplementary analysis including only terrestrial birds.

In the abstract you suggest that because density was positively affected by subsidies, whereas richness was negatively affected, some birds may benefit at the expense of others (i.e. competitive exclusion I assume). This is an interesting idea, but is not very well developed elsewhere in the paper. This could easily be examined in your study, to see which species, or families, tend to benefit and increase in density, and which drop out. It would add to the story if you could include a figure, and some discussion on this.

The influence of the subsidies on the patterns is weak. This should be made clearer in the abstract, including that island area is still dominating patterns.

In the discussion, on page 8, lines 299-311, none of the mechanisms are actually about the subsidy / D15N. They are about disturbance by otters, or changes to vegetation by otters and wind/marine spray. The manuscript misses a discussion of how the D15N results are likely influencing the birds - i.e. the mechanism of these differences in soil signatures driving bird richness and density.

From your description of your analyses in the methods, I can see how isolation was dropped from the richness models, but it is less clear what happens to isolation for density. I assume the same model evaluation process was carried out for density, including the potential to include isolation, but this is not clearly stated.

Page 4, line 163 - your 'island-wide measure' is unlikely, as the samples were taken from the edges, and the mechanisms you propose for the marine subsidies are very likely to be greatest around the edges and dissipate as you move into the island. This could be why you get stronger effects for small islands (edge to area ratio). This issue needs some discussion in the manuscript.

Page 4, line 147 - tell us the methods here, don't just refer the reader to Wickham et al.

Your results paragraphs give model estimated values for richness 1 SD above and below average size islands. For density you call this 1 order of magnitude above and below. This should be consistent.

Page 8, line 291 - this paragraph could be moved further down the discussion.

Finally, the writing could be more concise throughout the manuscript.

Decision letter (RSPB-2019-1889.R0)

12-Nov-2019

Dear Miss Obrist:

I am writing to inform you that your manuscript RSPB-2019-1889 entitled "Marine subsidies drive

patterns in avian island biogeography" has, in its current form, been rejected for publication in Proceedings B.

This action has been taken on the advice of referees, who have recommended that substantial revisions are necessary. With this in mind we would be happy to consider a resubmission, provided the comments of the referees are fully addressed. However please note that this is not a provisional acceptance.

Sincerely,

Dr Daniel Costa
mailto:proceedingsb@royalsociety.org

Reviewer(s)' Comments to Author:

Referee: 1

Comments to the Author(s)

This is an interesting and well-written paper which tests a hypothesis on a new twist to the classical equilibrium theory of island biogeography; the effect of marine subsidies on the species area curve. The hypothesis incorporates the proposed hump-shaped relationship between species richness and productivity, making the idea captivating. Previous studies of plants have found evidence for this hypothesis, but similar studies of mobile animals to do exist to my knowledge, making this study unique. The research team is highly qualified for this investigation and their analytical methods are superb. The results which provide some support for the hypothesis are intriguing; terrestrial bird species richness decreased with increasing marine subsidies. However, the conclusion on the mechanism of this effect is quite tentative for several reasons.

The basic premise is that productivity of the study system is high which is on the declining side of diversity-productivity curve. Although previous studies have found that productivity is high on the mainland of British Columbia, productivity on the small study islands may have been substantially lower, and thereby on the increasing side of the diversity-productivity curve. If so, according to the hypothesis, the relationship between species richness and marine subsidies should be positive, not negative as found in this study.

Previous studies cited in this paper on the mainland of British Columbia found that salmon-derived nutrients increased diversity of terrestrial bird species (Field, R.D. & Reynolds, J.D. 2011, Wagner, M.A. & Reynolds, J.D. 2019), but no explanation is given for the conflicting results of those studies and this study. This needs to be addressed.

In accordance with the subsidized island biogeography theory, the authors suggest that increasing productivity via marine subsidies increased overall bird density but decreased species richness due to competitive exclusion by dominant species. However, there is no evidence for competition in the paper. Data on species presence/absence and abundances on islands with high and low subsidies may shed some light on competitive interactions. As it stands now, support for the subsidized island biogeography theory is weak. I must commend the authors for including alternative hypotheses. Predators associated with marine subsidies, such as otters, eagles and sea birds (particularly seagulls) may exclude certain highly vulnerable terrestrial bird species but not others that are less vulnerable to predation. The other alternative hypothesis is that high inputs of marine subsidies may occur on islands with habitats that are not suitable for some bird species. This phenomenon may be common in other systems as well. Data on the association between habitat types and marine subsidies would be helpful. I feel that the alternative hypotheses are quite plausible and deserve more attention.

The Abstract and Results state the positive effect of subsidies on density is stronger on smaller islands but there is no reference to a test for this pattern. Is it the interaction between area and forest edge subsidies? A plot of area vs forest edge subsidies would be helpful.

Referee: 2

Comments to the Author(s)

Obrist et al. assess how marine subsidies influence island biogeography patterns in birds, with a well replicated study in western Canada. This is a nice contribution, and will add a robust data-rich example to this debate. I do have some issues for the authors to address.

It is not clear enough in the manuscript if your response (the bird data) are all terrestrial bird species, or if it includes some marine birds. In the methods you note that seabird guano is one of the contributing inputs for the marine subsidies. It is important to be clear if your response is just terrestrial birds, that feed on land, to remove the potential for circularity in the narrative / analysis (i.e. the bird numbers meaning more marine D15N signal in the soil). If your response does include both terrestrial and marine birds, I suggest you do a supplementary analysis including only terrestrial birds.

In the abstract you suggest that because density was positively affected by subsidies, whereas richness was negatively affected, some birds may benefit at the expense of others (i.e. competitive exclusion I assume). This is an interesting idea, but is not very well developed elsewhere in the paper. This could easily be examined in your study, to see which species, or families, tend to benefit and increase in density, and which drop out. It would add to the story if you could include a figure, and some discussion on this.

The influence of the subsidies on the patterns is weak. This should be made clearer in the abstract, including that island area is still dominating patterns.

In the discussion, on page 8, lines 299-311, none of the mechanisms are actually about the subsidy / D15N. They are about disturbance by otters, or changes to vegetation by otters and wind/marine spray. The manuscript misses a discussion of how the D15N results are likely influencing the birds - i.e. the mechanism of these differences in soil signatures driving bird richness and density.

From your description of your analyses in the methods, I can see how isolation was dropped from the richness models, but it is less clear what happens to isolation for density. I assume the same model evaluation process was carried out for density, including the potential to include isolation, but this is not clearly stated.

Page 4, line 163 – your ‘island-wide measure’ is unlikely, as the samples were taken from the edges, and the mechanisms you propose for the marine subsidies are very likely to be greatest around the edges and dissipate as you move into the island. This could be why you get stronger effects for small islands (edge to area ratio). This issue needs some discussion in the manuscript.

Page 4, line 147 – tell us the methods here, don’t just refer the reader to Wickham et al.

Your results paragraphs give model estimated values for richness 1 SD above and below average size islands. For density you call this 1 order of magnitude above and below. This should be consistent.

Page 8, line 291 – this paragraph could be moved further down the discussion.

Finally, the writing could be more concise throughout the manuscript.

Author's Response to Decision Letter for (RSPB-2019-1889.R0)

See Appendix A.

RSPB-2020-0108.R0

Review form: Reviewer 1

Recommendation

Accept with minor revision (please list in comments)

Scientific importance: Is the manuscript an original and important contribution to its field?

Good

General interest: Is the paper of sufficient general interest?

Good

Quality of the paper: Is the overall quality of the paper suitable?

Good

Is the length of the paper justified?

Yes

Should the paper be seen by a specialist statistical reviewer?

No

Do you have any concerns about statistical analyses in this paper? If so, please specify them explicitly in your report.

No

It is a condition of publication that authors make their supporting data, code and materials available - either as supplementary material or hosted in an external repository. Please rate, if applicable, the supporting data on the following criteria.

Is it accessible?

Yes

Is it clear?

Yes

Is it adequate?

Yes

Do you have any ethical concerns with this paper?

No

Comments to the Author

I am satisfied with the authors' response to my comments, but have some minor comments on the revision.

Abstract line 59: change positive to negative. Note that the coefficient is negative in fig. 2f, indicating that the effect of marine subsidy decreases with island area.

Line 176: insert "negative" before "correlation".

Line 185: move (120,245 m²) to after "nearest island large enough".

Line 240: Include "The interaction coefficient was negative indicating that this marine subsidy effect decreased with island area."

Supplementary Information. Figure 4S legend: change {Wilman, 2014 #404} to (Wilman et al. 2004). References [4]: insert "Ecology" before 95.

Review form: Reviewer 2

Recommendation

Accept as is

Scientific importance: Is the manuscript an original and important contribution to its field?

Good

General interest: Is the paper of sufficient general interest?

Good

Quality of the paper: Is the overall quality of the paper suitable?

Good

Is the length of the paper justified?

Yes

Should the paper be seen by a specialist statistical reviewer?

No

Do you have any concerns about statistical analyses in this paper? If so, please specify them explicitly in your report.

No

It is a condition of publication that authors make their supporting data, code and materials available - either as supplementary material or hosted in an external repository. Please rate, if applicable, the supporting data on the following criteria.

Is it accessible?

Yes

Is it clear?

Yes

Is it adequate?

Yes

Do you have any ethical concerns with this paper?

No

Comments to the Author

None

Decision letter (RSPB-2020-0108.R0)

11-Feb-2020

Dear Miss Obrist

I am pleased to inform you that your manuscript RSPB-2020-0108 entitled "Marine subsidies mediate patterns in avian island biogeography" has been accepted for publication in Proceedings B.

The referee(s) have recommended publication, but also suggest some minor revisions to your manuscript. Therefore, I invite you to respond to the referee(s)' comments and revise your manuscript. Because the schedule for publication is very tight, it is a condition of publication that you submit the revised version of your manuscript within 7 days. If you do not think you will be able to meet this date please let us know.

[http://datadryad.org/submit?journalID=RSPB&manu=\(Document not available\)](http://datadryad.org/submit?journalID=RSPB&manu=(Document%20not%20available)) which will take you to your unique entry in the Dryad repository. If you have already submitted your data to dryad you can make any necessary revisions to your dataset by following the above link. Please see <https://royalsociety.org/journals/ethics-policies/data-sharing-mining/> for more details.

Sincerely,

Dr Daniel Costa
mailto: proceedingsb@royalsociety.org

Associate Editor
Board Member

Comments to Author:

Both reviewers and I are happy with this manuscript now. Reviewer two only suggests correcting a small number of minor wording corrections / typographical errors. Congratulations, this will be a great paper!

Reviewer(s)' Comments to Author:

Referee: 2

Comments to the Author(s).
None

Referee: 1

Comments to the Author(s).

I am satisfied with the authors' response to my comments, but have some minor comments on the revision.

Abstract line 59: change positive to negative. Note that the coefficient is negative in fig. 2f, indicating that the effect of marine subsidy decreases with island area.

Line 176: insert "negative" before "correlation".

Line 185: move (120,245 m²) to after "nearest island large enough".

Line 240: Include "The interaction coefficient was negative indicating that this marine subsidy effect decreased with island area."

Supplementary Information. Figure 4S legend: change {Wilman, 2014 #404} to (Wilman et al. 2004). References [4]: insert "Ecology" before 95.

Decision letter (RSPB-2020-0108.R1)

17-Feb-2020

Dear Miss Obrist

I am pleased to inform you that your manuscript entitled "Marine subsidies mediate patterns in avian island biogeography" has been accepted for publication in Proceedings B.

You can expect to receive a proof of your article from our Production office in due course, please check your spam filter if you do not receive it. PLEASE NOTE: you will be given the exact page

length of your paper which may be different from the estimation from Editorial and you may be asked to reduce your paper if it goes over the 10 page limit.

Your article has been estimated as being 8 pages long. Our Production Office will be able to confirm the exact length at proof stage.

Open Access

Paper charges

Sincerely,

Proceedings B

Appendix A

Response to referees

Referee: 1

Comment 1:

This is an interesting and well-written paper which tests a hypothesis on a new twist to the classical equilibrium theory of island biogeography; the effect of marine subsidies on the species area curve. The hypothesis incorporates the proposed hump-shaped relationship between species richness and productivity, making the idea captivating. Previous studies of plants have found evidence for this hypothesis, but similar studies of mobile animals do not exist to my knowledge, making this study unique. The research team is highly qualified for this investigation and their analytical methods are superb. The results which provide some support for the hypothesis are intriguing; terrestrial bird species richness decreased with increasing marine subsidies. However, the conclusion on the mechanism of this effect is quite tentative for several reasons.

The basic premise is that productivity of the study system is high which is on the declining side of diversity-productivity curve. Although previous studies have found that productivity is high on the mainland of British Columbia, productivity on the small study islands may have been substantially lower, and thereby on the increasing side of the diversity-productivity curve. If so, according to the hypothesis, the relationship between species richness and marine subsidies should be positive, not negative as found in this study.

Previous studies cited in this paper on the mainland of British Columbia found that salmon-derived nutrients increased diversity of terrestrial bird species (Field, R.D. & Reynolds, J.D. 2011, Wagner, M.A. & Reynolds, J.D. 2019), but no explanation is given for the conflicting results of those studies and this study. This needs to be addressed.

Response to comment 1:

This is a good point, and one that we also considered extensively when we first obtained our results. Because of the unimodal diversity-productivity curve, we were aware of the possibility that we *could* see a negative relationship with increased subsidies. Given that our islands are likely even more nutrient-limited than the coastal waterways in those mainland studies, however, we thought it was unlikely that productivity would be high enough to cause declines in species richness.

Overall, we think the main reason that we are seeing conflicting results with previous studies is due to the source of the subsidy, and we have added lines 280 - 290 to discuss this in our manuscript. Essentially, salmon provide nutrients in a predictable manner, without disturbance to coastal vegetation, and without increasing predation risk to passerines. Marine input from river otter activity, on the other hand, is extremely variable across time and space. Otters are gregarious and extremely mobile. They feed on fish, marine invertebrates, aquatic invertebrates, and amphibians, as well as on some bird species (mostly waterfowl). Although there is no published evidence that they eat songbirds, anecdotally, we did see at least one songbird carcass in an otter scat during this project. It is possible that birds avoid more “ottery” areas out of fear of predation. Otters can also cause major habitat disturbances, which has been shown to have

negative effects on bird species diversity, although we were not able to test this conclusively with our dataset.

Comment 2:

In accordance with the subsidized island biogeography theory, the authors suggest that increasing productivity via marine subsidies increased overall bird density but decreased species richness due to competitive exclusion by dominant species. However, there is no evidence for competition in the paper. Data on species presence/absence and abundances on islands with high and low subsidies may shed some light on competitive interactions. As it stands now, support for the subsidized island biogeography theory is weak. I must commend the authors for including alternative hypotheses. Predators associated with marine subsidies, such as otters, eagles and sea birds (particularly seagulls) may exclude certain highly vulnerable terrestrial bird species but not others that are less vulnerable to predation. The other alternative hypothesis is that high inputs of marine subsidies may occur on islands with habitats that are not suitable for some bird species. This phenomenon may be common in other systems as well. Data on the association between habitat types and marine subsidies would be helpful. I feel that the alternative hypotheses are quite plausible and deserve more attention.

Response to comment 2:

In response to this comment (and Referee 2 Comment 2 below), we assessed whether certain guilds vary in predation vulnerability and/or have different habitat affinities by analysing species richness and population density by guild. We did not find evidence of competitive exclusion. It appears that the insectivore guild may be driving patterns, but there is a lot of uncertainty around estimates since most of these islands are so sparsely populated.

Changes we made based on this comment:

- 1.) Wording cleared up throughout manuscript that we don't think competitive exclusion is responsible for driving patterns, including Discussion line 263.
- 2.) Added lines 294 – 296 to Discussion about how insectivores may be driving patterns.
- 3.) Included figure on guild-level analysis in SI (Figure 4S). A brief explanation of how we conducted this analysis (e.g., how guilds were assigned) is included in the figure caption.

We also conducted a tolerance analysis to detect species-specific responses to the level of $\delta^{15}\text{N}$. We did not include this in the manuscript since we found no immediate pattern in species tolerance to $\delta^{15}\text{N}$ that cannot also be attributed to island size. A more sophisticated analysis of individual species' habitat affinities, traits, and co-occurrences is an important next step but would be beyond the scope of the current community-level analysis.

Overall, we agree with Referee 1 – given that the main mechanism for a decrease in species richness due to a subsidy in SIB is competitive exclusion, our study does not add support to that particular component of the theory. We have changed line 263 in the Discussion to reflect this.

Comment 3:

The Abstract and Results state the positive effect of subsidies on density is stronger on smaller islands but there is no reference to a test for this pattern. Is it the interaction between area and forest edge subsidies? A plot of area vs forest edge subsidies would be helpful.

Response to comment 3:

Yes, we had originally included several potential interactions in our analyses, including one between area and forest edge subsidies. We have now made this clearer in the Methods by adding lines 200 - 206. The interaction between area and forest edge subsidies was significant in the avian density model, and we describe it in lines 243 – 244. It is also included in our coefficient plot (Figure 2f). In response to this comment, we have now added a plot of area vs forest edge subsidies in the SI (Figure 3S).

Referee: 2**Comment 1:**

Obrist et al. assess how marine subsidies influence island biogeography patterns in birds, with a well replicated study in western Canada. This is a nice contribution, and will add a robust data-rich example to this debate. I do have some issues for the authors to address.

It is not clear enough in the manuscript if your response (the bird data) are all terrestrial bird species, or if it includes some marine birds. In the methods you note that seabird guano is one of the contributing inputs for the marine subsidies. It is important to be clear if your response is just terrestrial birds, that feed on land, to remove the potential for circularity in the narrative / analysis (i.e. the bird numbers meaning more marine D15N signal in the soil). If your response does include both terrestrial and marine birds, I suggest you do a supplementary analysis including only terrestrial birds.

Response to comment 1:

We only included terrestrial birds known to breed in the area – we checked the BC Breeding Bird Atlas when deciding which species to include in the analysis. We made this clearer by adding “terrestrial breeding birds” or just “terrestrial birds” instead of just “birds” in several places in the Abstract (line 52), Introduction (line 91 & 109), Methods (lines 140 & 141), and Results (lines 221 & 223), Discussion (lines 252 & 254).

Comment 2:

In the abstract you suggest that because density was positively affected by subsidies, whereas richness was negatively affected, some birds may benefit at the expense of others (i.e. competitive exclusion I assume). This is an interesting idea, but is not very well developed elsewhere in the paper. This could easily be examined in your study, to see which species, or

families, tend to benefit and increase in density, and which drop out. It would add to the story if you could include a figure, and some discussion on this.

Response to comment 2:

In response to this comment (and Referee 1 Comment 2 above), we assessed whether certain species/guilds are more or less vulnerable to predation, and/or have different habitat affinities by analysing species richness and population density by guild, and by conducting a species-level tolerance analysis. As explained above, we found no evidence of competitive exclusion in either way. It appears that the insectivore guild may be driving patterns, but there is a lot of uncertainty around estimates.

Changes we made based on this comment:

- 4.) Wording cleared up throughout manuscript that we don't think competitive exclusion is responsible for driving patterns, including Discussion line 263.
- 5.) Added lines 294 – 296 to Discussion about how insectivores may be important.
- 6.) Included figure on guild-level analysis in SI (Figure 4S).

Comment 3:

The influence of the subsidies on the patterns is weak. This should be made clearer in the abstract, including that island area is still dominating patterns.

Response to comment 3:

Done. We have changed the title from “Marine subsidies **drive** patterns in avian island biogeography” to “Marine subsidies **mediate** patterns in avian island biogeography” and adjusted the wording in the Abstract to make it clear that the effect of area is the strongest.

Comment 4:

In the discussion, on page 8, lines 299-311, none of the mechanisms are actually about the subsidy / D15N. They are about disturbance by otters, or changes to vegetation by otters and wind/marine spray. The manuscript misses a discussion of how the D15N results are likely influencing the birds – i.e. the mechanism of these differences in soil signatures driving bird richness and density.

Response to comment 4:

Done. In lines 282-283, we now discuss the actual mechanism of subsidy – i.e.) soils are enriched, stimulating plant productivity.

Comment 5:

From your description of your analyses in the methods, I can see how isolation was dropped from the richness models, but it is less clear what happens to isolation for density. I assume the

same model evaluation process was carried out for density, including the potential to include isolation, but this is not clearly stated.

Response to comment 5:

Yes, we used the same model evaluation process for the density model. We added lines 209 – 211 to state this more clearly. The results for these models are in the SI Table 4S. We have also added a reference to this table in the main text.

Comment 6:

Page 4, line 163 – your ‘island-wide measure’ is unlikely, as the samples were taken from the edges, and the mechanisms you propose for the marine subsidies are very likely to be greatest around the edges and dissipate as you move into the island. This could be why you get stronger effects for small islands (edge to area ratio). This issue needs some discussion in the manuscript.

Response to comment 6:

In both the Methods and Discussion we changed the wording from an “island-wide measure” to a “single measure for each island”.

It is correct that subsidies concentrate around the edges of islands, and therefore, smaller islands with higher perimeter-to-area ratios should experience higher per-unit-area effects of subsidies. This is part of the premise for Anderson and Wait’s (2001) subsidized theory of island biogeography. However, we wanted to standardize the amount of subsidy available, and so decided to only include the shoreline plots to limit the effect of the edge to area ratio.

We had assumed that island size would not matter for the amount of subsidies coming on to shore – however, we found that smaller islands had higher levels of $\delta^{15}\text{N}$ even at 4 forest plots directly adjacent to the shore. We have added a figure in the SI for the relationship between island size and soil $\delta^{15}\text{N}$ for our study islands (Figure 3S). We have also made this point clearer in lines 167 - 175 in the Methods section and 265 - 268 in the Discussion.

Comment 7:

Page 4, line 147 – tell us the methods here, don’t just refer the reader to Wickham et al.

Response to comment 7:

Done – the methods paragraph about wrack biomass methods (lines 147 – 153) now includes specific details about how data were collected.

Comment 8:

Your results paragraphs give model estimated values for richness 1 SD above and below average size islands. For density you call this 1 order of magnitude above and below. This should be consistent.

Response to comment 8:

Done – we now call it one order of magnitude in both the density and richness results paragraphs.

Comment 9:

Page 8, line 291 – this paragraph could be moved further down the discussion.

Response to comment 9:

We have moved this paragraph down to line 309, where it fits better.

Comment 10:

Finally, the writing could be more concise throughout the manuscript.

Response to comment 10:

We have made multiple adjustments throughout to tidy up the manuscript, decreasing our word count from 5665 to 5273.